# Dynamics of Strategy Distributions in a One-Dimensional Continuous Trait Space for Games with a Quadratic Payoff Function

**Georgiy Karev**

National Center for Biotechnology Information, National Institutes of Health, Bldg. 38A, 8600 Rockville Pike, Bethesda, MD 20894, USA; karev@ncbi.nlm.nih.gov

**Abstract:** Evolution of distribution of strategies in game theory is an interesting question that has been studied only for specific cases. Here I develop a general method to extend analysis of the evolution of continuous strategy distributions given a quadratic payoff function for any initial distribution in order to answer the following question—given the initial distribution of strategies in a game, how will it evolve over time? I look at several specific examples, including normal distribution on the entire line, normal truncated distribution, as well as exponential and uniform distributions. I show that in the case of a negative quadratic term of the payoff function, regardless of the initial distribution, the current distribution of strategies becomes normal, full or truncated, and it tends to a distribution concentrated in a single point so that the limit state of the population is monomorphic. In the case of a positive quadratic term, the limit state of the population may be dimorphic. The developed method can now be applied to a broad class of questions pertaining to evolution of strategies in games with different payoff functions and different initial distributions.

**Keywords:** continuous strategy space; quadratic payoff function; evolution of distribution; HKV method

## 1. Introduction

Game-theoretic approach to population dynamics developed by Maynard Smith [1,2] and many other authors (see, for example, Reference [3]) assumes that individual fitness results from payoffs received during pairwise interactions that depend on individual phenotypes or strategies.

The approach to studying strategy-dependent payoffs in the case of a finite number of strategies is as follows. Assume $\pi(x, y)$ is the payoff received by an individual using strategy $x$ against one using strategy $y$. If there is a finite number of possible strategies (or traits), then $\pi(x, y)$ is an entry of the payoff matrix. Alternatively, the number of strategies may belong to a continuous rather than discrete set of values. The case when individuals in the population use strategies that are parameterized by a single real variable $x$ that belongs to a closed and bounded interval $[a, b]$ was studied in [4–10] as well as many others. A brief survey of recent results on continuous state games can be found in Reference [6].

Specifically, the case of quadratic payoff function was considered in References [11,12] and some others.

Taylor and Jonker [13] offered a dynamical approach for game analysis known as replicator dynamics that allows tracing evolution of a distribution of individual strategies/traits. Typically, it is assumed that every individual uses one of finitely many possible strategies parameterized by real numbers; in this case, the Taylor-Jonker equation can be reduced to a system of differential equations and solved using well-developed methods, subject to practical limitations stemming from possible high dimensionality of the system.

Here, I extend the approach of studying games with strategies that are parameterized by a continuous set of values to study the evolution of strategy (trait) distributions over time. Specifically, I develop a method that allows computing the current distribution for games with quadratic, as well as several more general payoff, functions at any time and for any initial distribution. The approach is close to the HKV (after hidden keystone variables) method developed in References [14–16] used for modeling evolution of heterogeneous populations and communities. It allows generation of more general results than have previously been possible.

## 2. Results

### 2.1. Master Model

Consider a closed inhomogeneous population, where every individual is characterized by a qualitative trait (or strategy) $x \in X$, where $X \subseteq R$ is a subset of real numbers. $X$ can be a closed and bounded interval $[a, b]$, a positive set of real numbers $R^+$ or the total set of real numbers $R$. Parameter $x$ describes an individual's inherited invariant properties; it remains unchanged for any given individual but varies from one individual to another. The fitness (per capita reproduction rate) $F(t, x)$ of an individual depends on the strategy $x$ and on interactions with other individuals in the population.

Let $l(t, x)$ be population density at time $t$ with respect to strategy $x$; informally, $l(t, x)$ is the number of individuals that use $x$-strategy.

Assuming overlapping generations and smoothness of $l(t, x)$ in $t$ for each $x \in X$, the population dynamics can be described by the following general model:

$$
\begin{aligned}
\frac{dl(t,x)}{dt} &= l(t, x) F(t, x) \\
N(t) &= \int_X l(t, x) dx \\
P(t, x) &= \frac{l(t,x)}{N(t)}
\end{aligned}
\tag{1}
$$

where $N(t)$ is the total population size and $P(t, x)$ is the pdf of the strategy distribution at time $t$. The initial pdf $P(0, x)$ and the initial population size $N(0)$ are assumed to be given.

Let $\pi(x, y)$ be the payoff of an $x$-individual when it plays against a $y$-individual. Following standard assumptions of evolutionary game theory, assume that individual fitness $F(t, x)$ is equal to the expected payoff that the individual receives as a result of a random pairwise interaction with another individual in the population, that is,

$$
F(t, x) = \int_X \pi(x, y) P(t, y) dy.
\tag{2}
$$

Equations (1) and (2) make up the master model.

Here our main goal is to study the evolution of the pdf $P(t, x)$ over time. To this end, it is necessary to compute population density $l(t, x)$ and total population size $N(t)$, which will be done in the following section.

### 2.2. Evolution of Strategy Distribution in Games with Quadratic Payoff Function

Assume that the payoff $\pi(x, y)$ has the form

$$
\pi(x, y) = -ax^2 + bxy + cx + dy^2 + ey + f,
\tag{3}
$$

where $f = f(N)$ is the "background" fitness term that depends on the total population size $N$ but does not depend on individuals' traits and interactions; $a, b, c, d, e$ are constant coefficients.

Then

$$F(t,x) = \int_X \pi(x,y)P(t,y)dy = -ax^2 + bxE^t[x] + cx + dE^t\left[x^2\right] + eE^t[x] + f(N),\tag{4}$$

where expected value is notated as $E^t[g(x)] = \int_X g(x)P(t,x)dx$.

Now population dynamics is defined by the equation

$$\frac{dl(t,x)}{dt} = l(t,x)\left(-ax^2 + bxE^t[x] + cx + dE^t\left[x^2\right] + eE^t[x] + f(N)\right).$$

In order to solve this equation, apply the version of HKV method [14–16]. Introduce auxiliary variables $s(t)$, $h(t)$, such that

$$\frac{ds}{dt} = E^t[x],$$
$$\frac{dh}{st} = E^t\left[x^2\right] + f(N(t))\tag{5}$$
$$s(0) = h(0) = 0.$$

Then

$$l(t,x) = l(0,x)e^{(es(t)+dh(t))}e^{-atx^2+x(ct+bs(t))},\tag{6}$$

$$N(t) = \int_X l(t,x)dx = N(0)e^{(es(t)+dh(t))}\int_X e^{-atx^2+x(bs(t)+ct)}P(0,x)dx,\tag{7}$$

$$P(t,x) = \frac{l(t,x)}{N(t)} = P(0,x)\frac{e^{-atx^2+x(bs(t)+ct)}}{\int_X e^{-atx^2+x(bs(t)+ct)}P(0,x)dx}.\tag{8}$$

Notice that $P(t,x)$ depends neither on $h(t)$ nor on $c,f,e$. Therefore, if one is interested in the distribution of strategies and how it changes over time rather than the density of x-individuals, then one can replace the reproduction rate given by Equation (4) by the reproduction rate

$$F(t,x) = -ax^2 + bxE^t[x] + cx.\tag{9}$$

Equivalently, one can use the payment function (3) in a simplified form

$$\pi(x,y) = -ax^2 + bxy + cx.\tag{10}$$

The model (1) with payoff function (10) and reproduction rate (9) has the same distribution of strategies as model (1) with payoff (3) and reproduction rate (4).

Next, using Equation (8), one can write $E^t[x]$ in the form

$$E^t[x] = \int_X xP(t,x)dx = \int_X xe^{-atx^2+x(ct+bs(t))}P(0,x)dx \Big/ \int_X e^{-atx^2+x(ct+bs(t))}P(0,x)dx.\tag{11}$$

Now define the following function $\Phi(t,\lambda)$, such that

$$\Phi(t,\lambda) = \int_X e^{-atx^2+x\lambda}P(0,x)dx.\tag{12}$$

$E^t[x]$ can now be expressed as

$$E^t[x] = \frac{\partial\Phi(t,ct+bs(t))}{\partial\lambda}\Big/\Phi(t,ct+bs(t)).\tag{13}$$

It is now possible to write the explicit equation for the auxiliary variable as

$$\frac{ds}{dt} = E^t[x] = \frac{\partial ln\Phi(t,\lambda)}{\partial \lambda} \Big/_{\lambda=bs(t)+ct}. \tag{14}$$

Next,

$$E^t[x^2] = \frac{\partial^2 \Phi(t, bs(t) + ct)}{\partial \lambda^2} \Big/ \Phi(t, bs(t) + ct)$$

and therefore

$$Var^t[x] = \frac{\partial^2 \Phi(t, bs(t) + ct)}{\partial \lambda^2} \Big/ \Phi(t, bs(t) + ct) - \left(\frac{\partial \Phi(t, bs(t) + ct)}{\partial \lambda} \Big/ \Phi(t, bs(t) + ct)\right)^2. \tag{15}$$

The moment generation function (mgf) of the current distribution of strategies as given by Equation (8) is

$$M_t(\delta) = \frac{\int_X e^{-atx^2 + x(bs(t)+ct+\delta)} P(0,x)}{\int_X e^{-atx^2 + x(bs(t)+ct)} P(0,x) dx} = \frac{\Phi(t, bs(t) + ct + \delta)}{\Phi(t, bs(t) + ct)}. \tag{16}$$

Equations (8)–(16) now provide a tool for studying the evolution of the distribution of strategies of the quadratic payment model over time for any initial distribution.

## 2.3. Initial Normal Distribution

The evolution of normal distribution in games with the quadratic payoff function has already been mostly studied; as shown by Oechssier and Riedel [6,8] and Cressman and Hofbauer [5], the class of normal distributions is invariant with respect to replicator dynamics in games with quadratic payoff functions (3) with positive parameter $a$.

This statement immediately follows from Equation (8) for the current distribution of traits. Additionally, the class of normal distributions truncated in a (finite or infinite) interval $[a, b]$ is also invariant, see Section 2.6 for details and examples.

Now consider the dynamics of initial normal distributions in detail.

Let the initial distribution be normal with the mean $m$ and variance $\sigma^2$,

$$P(0,x) = \frac{1}{\sqrt{2\pi\sigma^2}} \exp\left(-\frac{(x-m)^2}{2\sigma^2}\right), \quad \infty < x < \infty; \tag{17}$$

Its mgf is given by

$$M[\delta] = \exp(\delta m + \frac{\delta^2 \sigma^2}{2}) \tag{18}$$

Denoting for brevity $\gamma = \frac{1}{2\sigma^2}$, one can compute the function $\Phi(t, \lambda)$ :

$$\Phi(t,\lambda) = \int_{-\infty}^{\infty} e^{-ax^2t + x\lambda} P(0,x) dx = \sqrt{\gamma/\pi} \int_{-\infty}^{\infty} e^{-x^2 at + x\lambda - \gamma(x-m)^2} dx = \sqrt{\frac{\gamma}{\gamma+at}} \exp\left(\frac{\lambda^2 + 4\gamma\lambda m - 4a\gamma m^2 t}{4(\gamma+at)}\right). \tag{19}$$

Next,

$$\frac{\partial \Phi(t,\lambda)}{\partial \lambda} = \sqrt{\frac{\gamma}{\gamma+at}} \frac{2\gamma m + \lambda}{2(\gamma+at)} \exp\left(\frac{\lambda^2 + 4\gamma\lambda m - 4a\gamma m^2 t}{4(\gamma+at)}\right),$$

So

$$\frac{\partial \Phi(t,\lambda)}{\partial \lambda} \Big/ \Phi(t,\lambda) = \frac{\lambda + 2\gamma m}{2(\gamma+at)}. \tag{20}$$

Then, according to Equation (5), the following explicit equation for auxiliary keystone variable emerges:

$$\frac{ds}{dt} = \frac{bs + ct + 2\gamma m}{2(\gamma+at)}, \quad s(0) = 0. \tag{21}$$

This equation can be solved analytically as follows:

$$s(t) = \frac{c}{b(2a-b)}\left(bt + 2\gamma\left(1 - (1 + \frac{at}{\gamma})^{\frac{b}{2a}}\right) - \frac{2m\gamma}{b}\left(1 - (1 + \frac{at}{\gamma})^{\frac{b}{2a}}\right)\right). \tag{22}$$

Now it is possible to compute the mean, variance, and current distribution of strategies using Equations (12)–(15). In the case of normal initial distribution, the simplest way to do so is to use Equation (16) for the current mgf.

Indeed, using formula (16) and after simple algebra, one can write the current mgf as

$$M_t(\delta) = \frac{\Phi(t, ct + bs(t) + \delta))}{\Phi(t, ct + bs(t)))} = \exp\left(\frac{\delta(\lambda + 2m\gamma)}{2(\gamma + at)} + \frac{\delta^2}{4(\gamma + at)}\right).$$

It is exactly the mgf of the normal distribution (18) with the mean $\frac{\lambda + 2m\gamma}{2(\gamma + at)}$ and variance $\frac{1}{2(\gamma + at)}$.

Remembering that $\lambda = bs(t) + ct$ and using Equation (22), after some algebra the mean of the current strategy distribution takes the form

$$E^t[x] = (1 + \frac{at}{\gamma})^{\frac{b}{2a} - 1}(m - \frac{c}{2a - b}) + \frac{c}{2a - b}. \tag{23}$$

**Proposition 1.** *Let the initial distribution of strategies in model (1), (9) be normal $N(m, \sigma^2)$. Then the distribution of strategies at any time t is normal with the mean $E^t[x]$ given by Equation (23) and variance $Var^t[x] = \frac{1}{2(\gamma + at)} = \frac{\sigma^2}{1 + 2at\sigma^2}$.*

It is easy to see that if $2a - b > 0$, then $(1 + \frac{at}{\gamma})^{\frac{b}{2a} - 1} \to 0$ and $E^t[x] \to \frac{c}{2a - b}$ as $t \to \infty$; if $2a - b < 0$, then $(1 + \frac{at}{\gamma})^{\frac{b}{2a} - 1} \to \infty$; therefore $E^t[x] \to \infty$ if $m > \frac{c}{2a - b}$ and $E^t[x] \to -\infty$ if $m < \frac{c}{2a - b}$ as $t \to \infty$.

Notice that $E^t[x] \to m + \frac{c}{2a}\ln(1 + \frac{at}{\gamma})$ as $b \to 2a$, so $E^t[x] \to \infty$ if $2a - b \le 0$..

Figure 1 shows the dynamics of the mean of current distribution of traits.

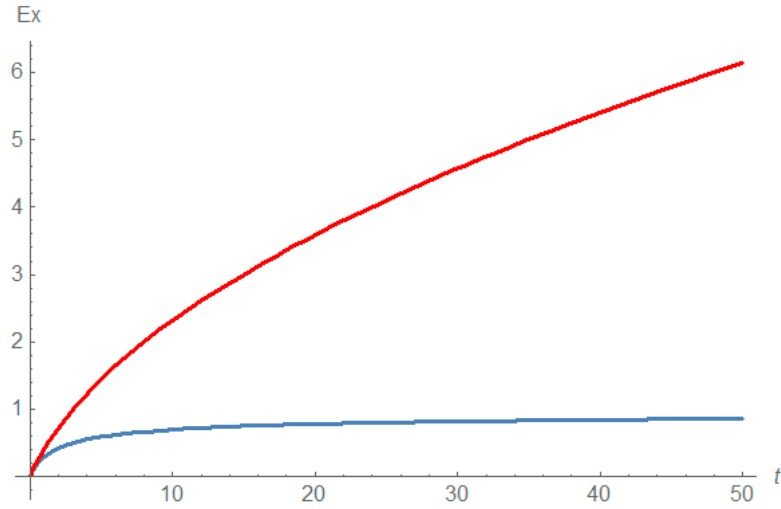

**Figure 1.** Dynamics of the mean value of current strategy distribution given by Equation (23) as $m = 0$; $b = 3$ (red), $b = 1$ (blue); other parameters: $a = c = \gamma = 1$. $E^t[x] \to \infty$ when $2a - b \le 0$; $E^t[x] \to \frac{c}{2a - b}$ when $2a - b > 0$.

Figure 2 shows the evolution of the distribution of traits over time. The variance of the current distribution $Var^t[x] = \frac{\sigma^2}{1+2at\sigma^2}$ tends to 0; therefore, the distribution of traits over time tends to a distribution concentrated at the point $x = \frac{c}{2a-b}$ for $2a - b > 0$.

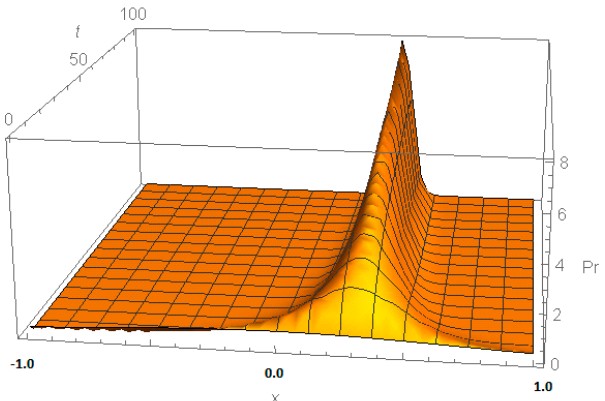

**Figure 2.** Evolution of the pdf $P(t, x)$ as given by Equation (22). The initial distribution is normal with $m = 0$, $\sigma^2 = 1$; parameters of the model are $a = 2$, $b = 1$, $c = 1$.

*2.4. Exponential Initial Distributions*

Let the initial distribution be exponential in $[0, \infty)$, $P(0, x) = ve^{-vx}$. Then

$$P(t, x) = \frac{e^{-ax^2 t + x(\lambda - v)}}{\int_0^\infty e^{-ax^2 t + x(\lambda - v)} dx} = 2\sqrt{\frac{at}{\pi}} \frac{e^{-at(x - \frac{\lambda - v}{2at})^2}}{1 + Erf\left[\frac{\lambda - v}{2\sqrt{at}}\right]}, \tag{24}$$

where $\lambda = bs(t) + ct$.

Equation (24) for any $t > 0$ describes the density of the normal distribution with the mean $m(t) = \frac{\lambda - v}{2at} = \frac{bs(t) + ct - v}{2at}$ and variance $\sigma^2(t) = \frac{1}{2at}$ truncated on $[0, \infty)$. Notably, the mean of the truncated normal distribution (24) is not equal to $m(t)$, and its variance is not equal to $\sigma^2(t)$. Instead, the mean of distribution (24) is

$$E^t[x] = m(t) + \frac{e^{-\frac{(ct + bs(t) - v)^2}{4at}}}{\sqrt{\pi at}(1 + Erf(\frac{ct + bs(t) - v}{2\sqrt{at}}))}. \tag{25}$$

In order to compute the mean given by Equation (25) and the current distribution (24) as a function of time one needs to solve for the auxiliary variable $s(t)$ that can be done using the function $\Phi(t, \lambda)$:

$$\Phi(t, \lambda) = v\int_0^\infty e^{-ax^2 t + x\lambda - vx} dx = \frac{\sqrt{\pi}}{2\sqrt{at}}v(1 + Erf(\frac{\lambda - v}{2\sqrt{at}}))e^{\frac{(\lambda - v)^2}{4at}}. \tag{26}$$

Then, according to Equation (14),

$$\frac{ds}{st} = \frac{\partial ln\Phi(t, \lambda)}{\partial \lambda}\Big/_{\lambda = bs(t) + ct} = \frac{ct + bs(t) - v}{2at} + \frac{e^{-\frac{(ct + bs(t) - v)^2}{4at}}}{\sqrt{\pi at}(1 + Erf(\frac{ct + bs(t) - v}{2\sqrt{at}}))}, \quad s(0) = 0. \tag{27}$$

This equation can be solved numerically. Using the solution $s(t)$, we can compute the distribution (24) and all its moments.

It follows from Equation (25), that $lim\, E^t[x] = lim\, m(t) = \frac{c}{2a} + \frac{b}{2a}lim\frac{s(t)}{t}$ as $t \to \infty$. One can show that $lim\frac{s(t)}{t} = \frac{c}{2a-b}$, therefore $lim\, E^t[x] = \frac{c}{2a}(\frac{b}{2a-b} + 1)$. The variance of the current distribution

tends to 0, so the limit distribution tends to a distribution concentrated in the point $x = \frac{c}{2a}(\frac{b}{2a-b} + 1)$. This proves the following proposition.

**Proposition 2.** *Let the initial distribution of strategies be exponential. Then the current distribution is normal at any time $t > 0$ that tends to a distribution concentrated in the point $x = \frac{c}{2a}(\frac{b}{2a-b} + 1)$.*

An example of the dynamics of the current mean and variance is given on Figure 3. Figure 4 shows the dynamics of the initial exponential distribution that turns to a truncated normal distribution with its variance tending to 0. Therefore, the current distribution tends to a distribution concentrated in the point $lim E^t[x] = 1$ as $t \to \infty$.

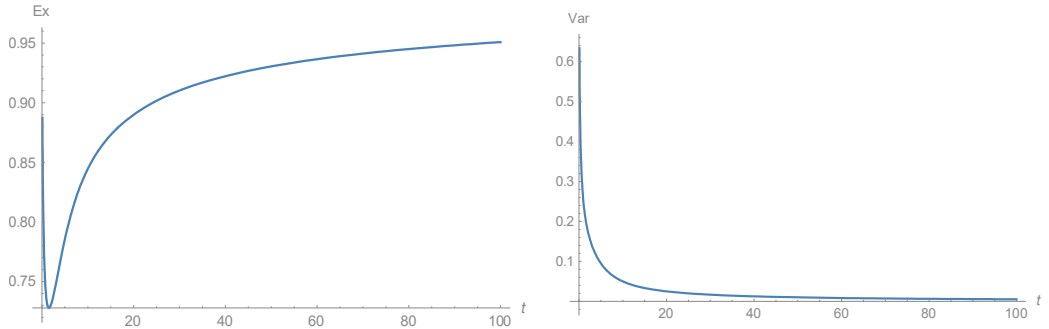

**Figure 3.** Plots of the mean (**left**) and variance (**right**) of distribution (24) with $a = b = c = v = 1$.

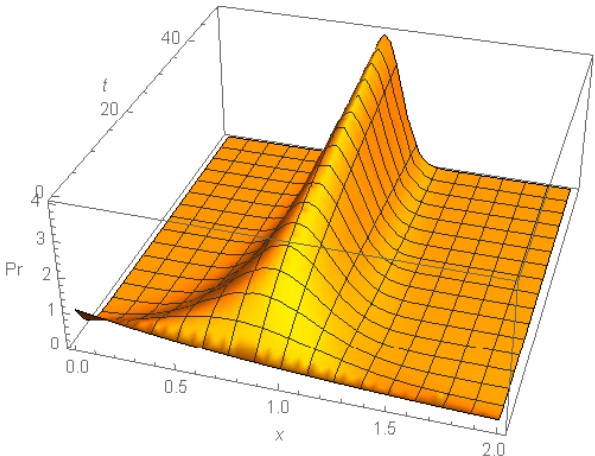

**Figure 4.** Evolution of the distribution of strategies over time given initial exponential distribution (24) with $a = b = c = v = 1$.

*2.5. Uniform Initial Distribution*

Now assume that the initial distribution is uniform in the interval $[-1, 1]$. Then

$$\Phi(t, \lambda) = \int_{-1}^{1} e^{-ax^2 t + x\lambda} dx = \frac{1}{2}\sqrt{\frac{\pi}{at}} \exp\left(\frac{\lambda^2}{4at}\right)\left(Erf\left(\frac{-\lambda + 2at}{2\sqrt{at}}\right) + Erf\left(\frac{\lambda + 2at}{2\sqrt{at}}\right)\right) \quad (28)$$

and the current distribution

$$P(t, x) = \frac{l(t, x)}{N(t)} = \frac{e^{-ax^2 t + x(ct + bs(t))}}{\Phi(t, ct + bs(t))}. \quad (29)$$

The auxiliary variable $s(t)$ can be computed using Equation (14), or, equivalently, directly using the expression (29) for the current pdf:

$$\frac{ds}{dt} = E^t[x] = \int_{-1}^{1} xP(t,x)dx = \frac{ct+bs(t)}{2at} + \frac{1}{\sqrt{\pi at}} \frac{\exp\left(-\frac{(bs(t)+ct+2at)^2}{4at}\right) - \exp\left(-\frac{(bs(t)+ct-2at)^2}{4at}\right)}{Erf\left(\frac{bs(t)+ct+2at}{2\sqrt{at}}\right) + Erf\left(\frac{-bs(t)-ct+2at}{2\sqrt{at}}\right)}. \tag{30}$$

For a positive parameter $a$, the distribution $P(t,x)$ is normal with the mean $E(t) = \frac{ct+bs(t)}{2at}$ and variance $\sigma^2(t) = \frac{1}{2at}$ truncated in the interval $[-1, 1]$. However, for negative values of parameter $a$ the distribution (29) is not normal; more specifically, if parameter $b$ is also negative, then the initial distribution evolves towards a U-shaped distribution, as can be seen Figure 5 (right).

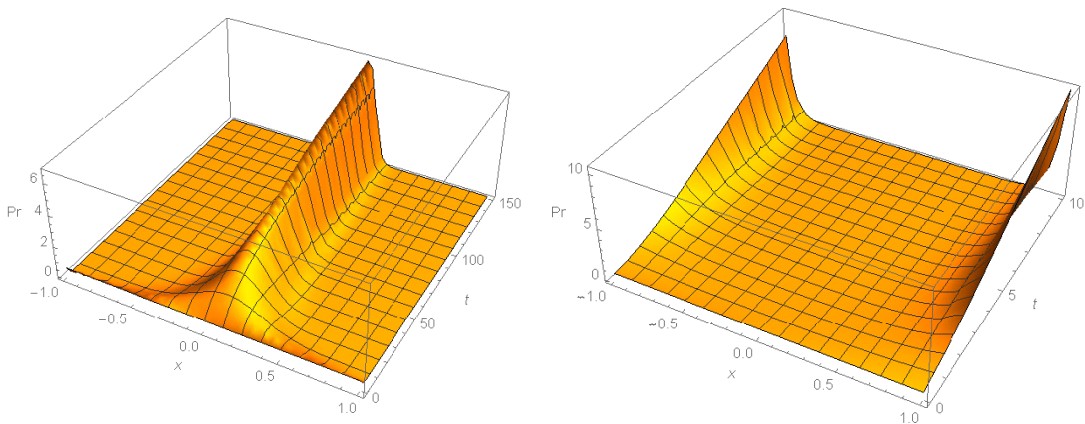

**Figure 5.** Evolution of the distribution of strategies over time given initial uniform distribution in $[-1, 1]$; left panel: $a = 1, b = -10, c = 1$; right panel: $a = -1, b = -10, c = 1$.

*2.6. Normal Initial Distribution Truncated in the Interval* $[-1, 1]$

Now assume the initial distribution is normal with zero mean, truncated in the interval $[-1, 1]$:

$$p(x) = Ce^{-(x/\sigma)^2}, -1 \le x \le 1, \tag{31}$$

with normalization constant $C = 1/\left[\sigma \sqrt{\pi} \, Erf\left(\frac{1}{\sigma}\right)\right]$.

Using the theory developed in Section 2.3, Equation (8), one can show that the current distribution of strategies is given by the formula

$$P(t,x) = \frac{2e^{-\frac{(bs(t)+ct-2(\gamma+at)x)^2}{4(\gamma+at)}}\sqrt{\gamma+at}}{\sqrt{\pi}\left(Erf\left[\frac{-bs(t)-ct+2(\gamma+at)}{2\sqrt{\gamma+at}}\right]+Erf\left[\frac{bs(t)+ct+2(\gamma+at)}{2\sqrt{\gamma+at}}\right]\right)} \text{ where } \gamma = 1/\sigma^2. \tag{32}$$

The distribution (32) is again normal truncated in the interval $[-1, 1]$. The current mean value that defines Equation (14) for the auxiliary variable $s(t)$ can be computed using Equation (13) or using the expression (32) for the current pdf. This way one can obtain a (rather bulky) equation for $s(t)$ that can be solved numerically. With this solution, one can trace the evolution of the initial truncated normal distribution. It can be shown that for $a > 0$ the variance of the current distribution tends to 0; therefore, the current distribution tends to a distribution concentrated in the point $limE^t[x]$ at $t \to \infty$. The value of $limE^t[x]$ depends on model parameters. Three examples of the evolution of strategy distribution are given in Figure 6.

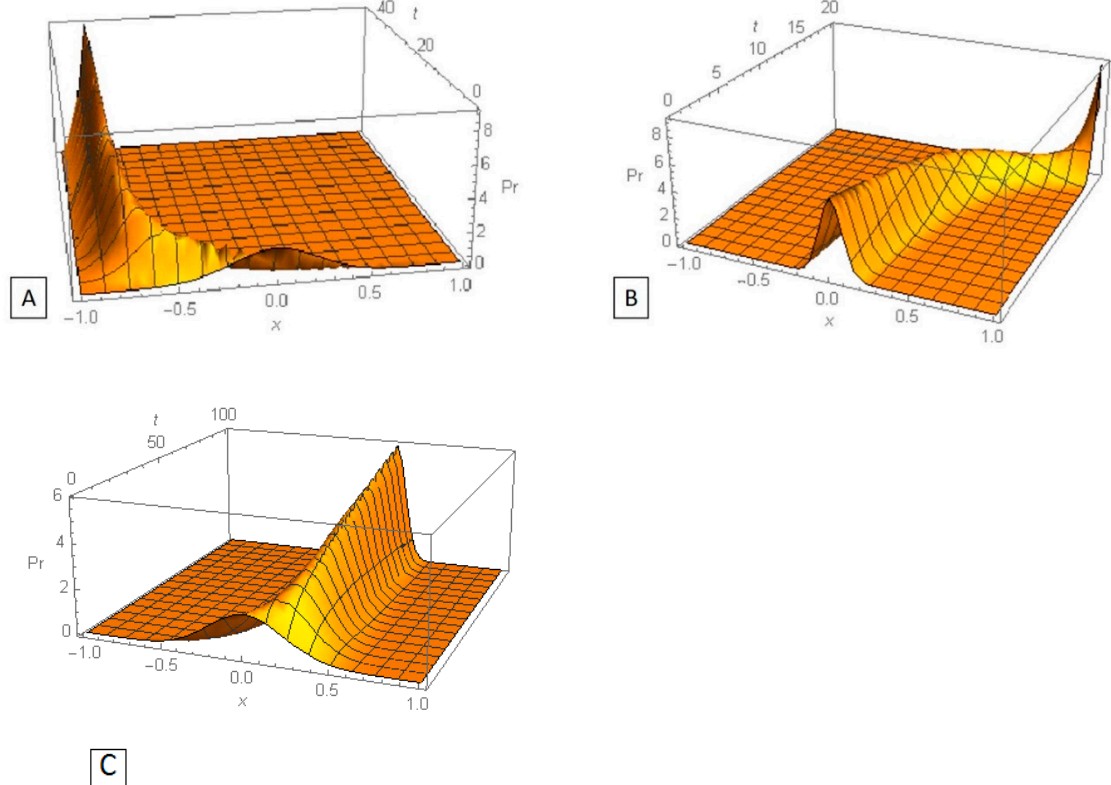

**Figure 6.** Evolution of the distribution of strategies over time given the initial truncated normal distribution. (**A**) $a = 5, b = -2, c = -10, \sigma^2 = 10$; (**B**) $a = -5, b = -2, c = 1, \gamma = 10$; (**C**) $a = 1, b = -2, c = 1, \gamma = 10$.

More generally, one can consider the normal distribution truncated in a finite interval $[a, b]$ or in a half-line $[a, \infty)$. Then it follows from Equation (8) that the current distribution is also normal truncated in that interval.

**Proposition 3.** *The class of truncated normal distributions is invariant with respect to replicator dynamics in games with quadratic payoff functions (3) with positive parameter a.*

In contrast, one can observe another kind of evolution of the initial truncated normal distribution for $a < 0$. Specifically, the current distribution has a U-shape and tends to a distribution concentrated in two extremal points of the interval where the initial distribution is defined, as can be seen in Figure 7.

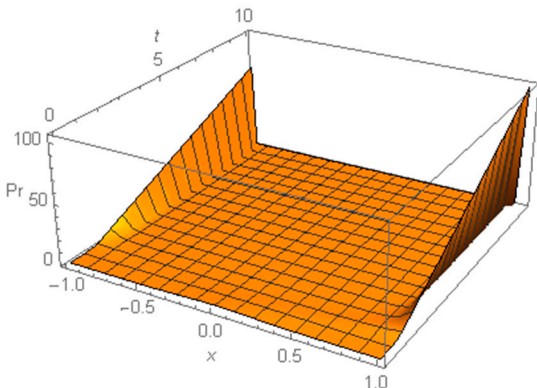

**Figure 7.** Evolution of the distribution of strategies over time given the initial normal distribution truncated in $[-1, 1]$; $a = -10$, $b = -6$, $c = 1$, $\gamma = 10$.

*2.7. Generalization*

The developed approach can be applied to a more general version of the payoff function:

$$\pi(x, y) = f_1(x) + f_2(x)f_3(y) + f_4(y). \tag{33}$$

In this case

$$\frac{dl(t, x)}{dt} = l(t, x) \int_X (f_1(x) + f_2(x)f_3(y) + f_4(y))P(t, y)dy = l(t, x)F(t, x),$$

where $F(t, x) = f_1(x) + f_2(x)E^t[f_3] + E^t[f_4]$.

Let us introduce auxiliary variables

$$\frac{ds}{dt} = E^t[f_3], \quad \frac{dh}{st} = E^t[f_4], \quad s(0) = h(0) = 0. \tag{34}$$

Then

$$l(t, x) = l(0, x) \exp[f_1(x)t + f_2(x)s(t) + h(t)],$$
$$N(t) = \int_X l(t, x)dx = N(0)e^{h(t)} \int_X e^{f_1(x)t + f_2(x)s(t)} P(0, x)dx, \tag{35}$$
$$P(t, x) = \frac{l(t, x)}{N(t)} = P(0, x) \frac{e^{f_1(x)t + f_2(x)s(t)}}{\int_X e^{f_1(x)t + f_2(x)s(t)} P(0, x)dx}.$$

One can see that the pdf $P(t, x)$ does not depend on the variable $h(t)$ and hence on the function $f_4(y)$. It follows from (35) that

$$E^t[f_3] = \int_x f_3(x)P(t, x)dx = \int_x f_3(x)e^{f_1(x)t + f_2(x)s(t)}P(0, x) / \int_X e^{f_1(x)t + f_2(x)s(t)}P(0, x)dx.$$

Then the equation

$$\frac{ds}{dt} = E^t[f_3], \quad s(0) = 0 \tag{36}$$

can be solved, at least numerically.

Another equivalent approach may also be useful. Define the function

$$\Phi(t, \lambda, \delta) = \int_x e^{f_1(x)t + \delta f_2(x) + \lambda f_3(x)} P(0, x)dx. \tag{37}$$

Then

$$E^t[f_3] = \frac{\partial ln\Phi(t, \lambda, \delta)}{\partial \lambda} /_{\lambda=0, \; \delta=s(t)}.$$

This results in a closed equation for the auxiliary variable $s(t)$ :

$$\frac{ds}{st} = E^t[f_2] = \frac{\partial ln\Phi(t, \lambda, \delta)}{\partial \lambda} /_{\lambda=0, \; \delta=t+cs(t)}. \tag{38}$$

Having the solution to equations (36) or (38), one can compute the current pdf (35) and all statistical characteristics of interest, such as the current mean and variance of strategies given any initial distribution.

**Example 1 (see [12], Example 1).** *Let* $\pi(x, y) = -ax^4 + 4xy$. *Then* $F(t, x) = -ax^4 + 4xE^t[x]$.

Introduce the auxiliary variable using the equation $\frac{ds}{dt} = E^t[x]$. Then

$$l(t, x) = l(0, x) \exp\left(-ax^4 t + 4xs(t)\right),$$
$$N(t) = \int_x l(t, x)dx = N(0) \int_X \exp\left(-ax^4 t + 4xs(t)\right)P(0, x)dx, \tag{39}$$
$$P(t, x) = \frac{l(t,x)}{N(t)} = P(0, x) \frac{\exp\left(-ax^4 t + 4xs(t)\right)}{\int_X \exp\left(-ax^4 t + 4xs(t)\right)P(0,x)dx}.$$

Let $\Phi(t, \lambda) = \int_x e^{-ax^4 t + \lambda x} P(0, x)dx$.
Then

$$\frac{ds}{dt} = E^t[x] = \frac{\partial ln\Phi(t, \lambda)}{\partial \lambda} /_{\lambda = 4s(t)}.$$

This equation can be solved numerically, allowing one to then compute the pdf according to Equation (39).

### 3. Discussion

Classical problems of evolutionary game theory are concentrated on studying equilibrium states (such as evolutionarily stable states and Nash equilibria). Notably, it takes indefinite time to reach any equilibrium when starting from a from non-trivial initial distribution of strategies in continuous-time models. Therefore, the evolution of a given initial distribution over time may be of great interest and potentially critical importance for studying real population dynamics.

Here I developed a method that allows extending the analysis of evolution of continuous strategy distributions in games with a quadratic payoff function. Specifically, the method described here allows us to answer the question: given an initial distribution of strategies in a game, how will it evolve over time? Typically, the dynamics of population distributions are governed by replicator equations, which appear both in evolutionary game theory, as well as in analysis of the dynamics of non-homogeneous populations and communities. The approach suggested here is based on the HKV (hidden keystone variable) method developed in References [9–11] for analysis of the dynamics of inhomogeneous populations and finding solutions of corresponding replicator equations. The method allows the computing of the current strategy distribution and all statistical characteristics of interest, such as current mean and variance, of the current distribution given any initial distribution at any time.

I looked at several specific examples of initial distributions:

o    Normal
o    Exponential
o    Uniform on [−1, 1]
o    Truncated normal on [−1, 1]

Through the application of the proposed method, I confirm the existing results given in References [5,6], that the family of normal distributions is invariant in a game with a quadratic payoff function with negative quadratic term. Additionally, I derive explicit formulas for the current distribution, its mean and variance. I show also that the class of truncated normal distributions is also invariant with respect to replicator dynamics in games with quadratic payoff functions; as an example, I consider in detail the case of initial normal distribution truncated in [−1, 1].

Notably and unexpectedly, in most cases, regardless of initial distribution, the current distribution of strategies in games with negative quadratic term is normal, standard or truncated. Over time it evolves towards a distribution concentrated in a single point that is equal to the limit values of the mean of the current normal distribution. This can have implications for a broad class of questions pertaining to evolution of strategies in games.

For instance, the question of whether the limit state of the population is mono - or polymorphic was discussed in the literature. Here I show that for games with a quadratic payoff function, the population tends to a monomorphic stable state if the quadratic term is negative. In contrast, if the quadratic term

of the payoff function is positive and the initial distribution is concentrated in a finite interval, then the current distribution can have a U-shape, and then the population tends to a di-morphic state.

In the last section I extend the developed approach to games with payoff functions of the form $\pi(x, y) = f_1(x) + f_2(x)f_3(y) + f_4(y)$. Formally, this framework can be applied to a very broad class of payoff functions, which include exponential or polynomial payoff functions; however, in many cases finding a solution to the equation for the auxiliary variable can be a difficult computational problem.

To summarize, the proposed method is validated against previously published results, and is then applied to a previously unsolvable class of problems. Application of this method could help expand the class of questions and answers that can now be obtained for a large class of problems in evolutionary game theory.

**Funding:** This research received no external funding.

**Conflicts of Interest:** The authors declare no conflict of interest.

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
