# Peer review of "Dynamics of Strategy Distributions in a One-Dimensional Continuous Trait Space for Games with a Quadratic Payoff Function"

_games, doi:10.3390/g11010014_

Round 1

Reviewer 1 Report

The paper „Dynamics of strategy distributions in a one-dimensional continuous trait space for games with quadratic payoff function“ gives a study of how an initial strategy distribution evolves over time.  Therefore, the stochastic ODEs describing the game dynamics are solved using a HKV approach known from modelling heterogeneous populations. The paper show the strategy dynamics for four different initial distributions, normal, exponential, uniform and truncated normal, partly confirming previous results. In general, I think this is an interesting paper that applies a mathematical technique from another domain to continuous game dynamics. I have three comments that should be taken into account to improve the overall significance of the paper. Also, extensive English edits are required. Particularly, the usage of definite and indefinite articles needs to be reviewed and corrected.   

(a) I think a major weakness of the paper is that there is no discussion about the game-theoretical implications of the findings, that is, the long-time dynamics of initial strategy distributions in the games you are considering. I recommend to add such a discussion as otherwise the paper might be mathematically interesting but of little relevance for the theory of games.  

(b) There is a substantial amount of recent works on games with continuous strategy space, which are mostly not covered in the references. From the references cited the impression could be formed that the topic is disregarded since the paper by Cressman and Hofbauer in 2005. This is not the case, see for instance D Hingu, KSM Rao, AJ Shaiju, Evolutionary stability of polymorphic population states in continuous games, Dynamic Games and Applications, 2018, or MW Cheung, Imitative dynamics for games with continuous strategy space, Games and Economic Behavior, 2016, or W Zhong, J Liu, L Zhang, Evolutionary dynamics of continuous strategy games on graphs and social networks under weak selection, Biosystems, 2013, or R Cressman, Y Tao, The replicator equation and other game dynamics, PNAS, 2014, to name just a few. I think recent works on the topic must be taken into account and the methodology and findings should be related to each other.

(c) The results are given for quadratic payoffs. Could you comment about your method for other types of payoff functions, for instance exponential payoff?

Author Response

Response to Reviewer 1

The paper „Dynamics of strategy distributions in a one-dimensional continuous trait space for games with “quadratic payoff function” gives a study of how an initial strategy distribution evolves over time.  Therefore, the stochastic ODEs describing the game dynamics are solved using a HKV approach known from modelling heterogeneous populations. The paper show the strategy dynamics for four different initial distributions, normal, exponential, uniform and truncated normal, partly confirming previous results. In general, I think this is an interesting paper that applies a mathematical technique from another domain to continuous game dynamics. I have three comments that should be taken into account to improve the overall significance of the paper. Also, extensive English edits are required. Particularly, the usage of definite and indefinite articles needs to be reviewed and corrected.    (a) I think a major weakness of the paper is that there is no discussion about the game-theoretical implications of the findings, that is, the long-time dynamics of initial strategy distributions in the games you are considering. I recommend to add such a discussion as otherwise the paper might be mathematically interesting but of little relevance for the theory of games.  

Author’s response:

I added an example (Figure 7) that together with Figure 6 shows that depending on the sign of the quadratic term of the payoff function the limit state of the population can be monomorphic or dimorphic. This question was discussed in the literature, which is now reflected in the Discussion section.

(b) There is a substantial amount of recent works on games with continuous strategy space, which are mostly not covered in the references. From the references cited the impression could be formed that the topic is disregarded.

Author’s response:

A list of references has been extended, and some recent papers on games with continuous strategy have now been added. I believe that a detailed survey of recent works on continuous state games is out of the scope of this manuscript because one can find it in the literature – see, for instance, a brief but substantial survey of recent results about continuous-state evolutionary games in D. Hingu et.al., 2018. The references have been updated in the text.

(c) The results are given for quadratic payoffs. Could you comment about your method for other types of payoff functions, for instance exponential payoff?

Author’s response:

In the last section of the revised manuscript I described how the developed approach can be applied to games with payoff functions of a general  form given by equation (8.1).

Formally, this framework includes exponential or polynomial payoff functions, but in these cases solving the equation with respect to the auxiliary variable may become a non-trivial computational problem. For this reason, I consider here only the quadratic payoff function, where the corresponding equation can be easily solved.  Additionally, I show in s.8 how the method can be applied for studying games with polynomial (degree 4) of payoff function, considered in [12].                                                             

I would like to thank the reviewer for the helpful comments.

Reviewer 2 Report

Authors develop a nice framework for analysis of continuous strategy games with quadratic payoff functions. The results focus on deriving time-dependent distributions for specific initial distributions (normal, exponential, uniform, truncated), and ends on a nice generalization of a generalized form of quadratic payoff.

Below, I offer a few minor suggestions to revise:

Page 1, line 25: shouldn’t Taylor-Jonker be a *non-linear* system of diff. eqns? Change from linear. Page 4, line 94-95. I’m not entirely clear what the author means by “statistical characteristics of the model.” This strikes me as an important result (that the model result does not depend on h(t), or c, f, e). The authors should add a few clarifying sentences to discuss the mathematical reason which this occurs, along with the the implications of this. The author should clarify the meaning “statistical characteristics,” (I assume they mean time-varying distribution of strategies, rather than the population density). Page 6, line 158: add a reference to eqn. 3.2 here, to aid the reader. Figure 1 needs a legend (and use different colors for values of b). What is the vertical axis of Figure 2? Is it P(t,x)? The scale is from 0 to 8, which confuses me (as P(t,x) should be bounded by 0 and 1, correct?). Perhaps this is plotting L(x,t)? Figure 5: I think it would be interesting to add a panel to Figure 5 which shows the simulation for a positive value of “a” for Figure 5. I think this would be informative and quite interesting. Page 11, line 239, this equation seems to have formatted incorrectly.

Author Response

Response to Reviewer 2

Authors develop a nice framework for analysis of continuous strategy games with quadratic payoff functions. The results focus on deriving time-dependent distributions for specific initial distributions (normal, exponential, uniform, truncated), and ends on a nice generalization of a generalized form of quadratic payoff.

Below, I offer a few minor suggestions to revise:

Page 1, line 25: shouldn’t Taylor-Jonker be a *non-linear* system of diff. eqns? Change from linear.

Author’s response:

Done; the word “linear” is deleted.

Page 4, line 94-95. I’m not entirely clear what the author means by “statistical characteristics of the model.” This strikes me as an important result (that the model result does not depend on h(t), or c, f, e). The authors should add a few clarifying sentences to discuss the mathematical reason which this occurs, along with the implications of this. The author should clarify the meaning “statistical characteristics,” (I assume they mean time-varying distribution of strategies, rather than the population density).

Author’s response:

Corrected; clarification was added.

Page 6, line 158: add a reference to eqn. 3.2 here, to aid the reader.

Author’s response:

Done. 

Figure 1 needs a legend (and use different colors for values of b).

Author’s response:

Done. 

What is the vertical axis of Figure 2? Is it P(t,x)? The scale is from 0 to 8, which confuses me (as P(t,x) should be bounded by 0 and 1, correct?). Perhaps this is plotting L(x,t)?

Author’s response:

The vertical axis of Figure 2 is the pdf P(t,x). It is not a probability but the probability density that may be arbitrarily large and even indefinite, e.g., in the case of the Dirac delta-function, which corresponds to a probability measure concentrated in a single point.

Figure 5: I think it would be interesting to add a panel to Figure 5 which shows the simulation for a positive value of “a” for Figure 5. I think this would be informative and quite interesting.

Author’s response:

The panel has been added.

Page 11, line 239, this equation seems to have formatted incorrectly.

Author’s response:

The equation is corrected.

I would very much like to thank the reviewer for thorough reading of the manuscript and helpful comments that improve clarity of the message.